# Synthesis of Gadolinium-Loaded Poly(N-vinyl-2-pyrrolidone) Nanogels Using Pulsed Electron Beam Ionizing Irradiation

**DOI:** 10.3390/polym17152100

**Published:** 2025-07-30

**Authors:** Nouria Bouchikhi, Aiysha Ashfaq, Mohamad Al-Sheikhly

**Affiliations:** 1Centre de Recherche Scientifique et Technique en Analyses Physico-Chimiques (CRAPC), Tipaza 42000, Algeria; nouria.bouchikhi21@gmail.com; 2Laboratoire de Recherche sur les Macromolecules (LRM), Universite Aboubekr Belkaïd de Tlemcen (UABT), Tlemcen 13000, Algeria; 3Department of Chemistry and Biochemistry, University of Maryland, College Park, MD 20742, USA; 4Department of Materials Science and Engineering, University of Maryland, College Park, MD 20742, USA

**Keywords:** nanohydrogels, ionizing radiation, contrast agent, gadolinium

## Abstract

Poly(N-vinyl-2-pyrrolidone), or PVP, nanogels loaded with gadolinium nitrate (Gd(NO_3_)_3_·6H_2_O) were synthesized by ionizing irradiation, aiming for potential applications in magnetic resonance imaging (MRI). A comprehensive characterization of PVP and Gd aqueous solutions with different VP-monomer-to-Gd ratios was conducted before and after irradiation. The results indicate a complexation between PVP and Gd ions before irradiation. The size of the nanogels exhibited a strong dependence on several factors, including PVP molecular weight, concentration, temperature, and the precise timing of Gd introduction relative to the irradiation process. A quantification study was conducted to investigate the impact of molecular weight, the VP/Gd ratio, and Gd addition before or after the irradiation process on the concentration of free Gd ions. These findings offer valuable insights into optimizing the synthesis of Gd-loaded PVP nanogels for potential applications, highlighting the critical factors that influence their size and stability.

## 1. Introduction

The integration of innovative technologies and approaches in cancer treatment is essential for advancing early detection and enabling cell-specific therapies. Magnetic resonance imaging (MRI) has become an indispensable diagnostic tool in recent decades, widely used in clinical settings for the detection and diagnosis of cancer. Its capability for deep tissue penetration and provision of high-resolution 3D spatial information, derived from signals generated by the relaxation of water hydrogen nuclei in response to magnetic fields, enables the creation of highly resolved anatomical images [1].

Despite its advantages, differentiating tumor regions from normal tissue in medical imaging, including MRI, remains a significant challenge. Improving MRI sensitivity to enhance tissue contrast is therefore a major focus of current research. One widely adopted strategy involves introducing suitable contrast agents (CAs) to accentuate differences between tissues and to yield more-detailed images. The two principal types of MRI contrast agents are (a) T1 or positive agents, which shorten proton longitudinal (spin–lattice) relaxation time; and (b) T2 or negative agents, which affect proton transverse (spin–spin) relaxation [2].

A variety of CAs are currently utilized to boost image contrast, including paramagnetic metal ions like lanthanides [3,4], iron oxide nanoparticles [5], and organic radical contrast agents [6]. Among these, gadolinium-based contrast agents (GdCAs) are extensively employed in MRI due to their ability to produce positive contrast (T1 CAs). They achieve this by influencing the relaxation rate (R1 = 1/T1) of water hydrogen nuclei surrounding the paramagnetic ions [2,7,8]. Their efficacy is further bolstered by gadolinium’s seven unpaired electrons, prolonged electronic relaxation, and substantial magnetic moments [9]. Conventionally, molecular gadolinium chelates, such as Gd-DTPA and Gd-DOTA, are commercially available and clinically used as T1 contrast agents [10]. However, these traditional GdCAs present several limitations, including short blood circulation times, rapid elimination from tissues, non-specific in vivo distribution, and the inherent toxicity of free Gd^3+^ ions.

To overcome these limitations, researchers have increasingly turned to nanotechnology to develop next-generation agents for simultaneous diagnosis and therapy. The encapsulation of Gd chelates in nanometric vectors has proven to be a promising approach. Advances in this area show that embedding gadolinium within or onto nanoparticles can substantially enhance the agents’ longitudinal relaxivity (r_1_), thereby improving MRI sensitivity [11,12,13]. In particular, recent studies have demonstrated that gadolinium-loaded nanogels not only improve MRI contrast but also offer potential for controlled anti-cancer drug delivery [14,15]. These promising results have sparked widespread interest in the development of nanogel-based systems that can serve both as contrast agents for MRI and as vectors for drug delivery.

Nanogels loaded with contrast agents can be prepared through several methods: (1) the spontaneous physical self-assembly of hydrophilic polymers [16,17]; (2) the polymerization of monomers within heterogeneous environments such as microemulsions [18]; and (3) the formation of covalent bonds between pre-existing polymer chains, leading to chemical crosslinking [19]. Careful selection of these methods is crucial, considering potential disadvantages associated with the use of organic solvents, surfactants, and impurities from these agents, which can complicate purification processes.

Among the techniques mentioned above, using ionizing radiation to induce chemical crosslinking of polymer chains offers exceptional practical advantages. This approach enables the synthesis of nanogels free from toxic monomers, crosslinking agents, or surfactants, thereby eliminating the need for purification steps and yielding a sterile product in a single synthesis step. Through the use of only polymer and water as substrates, the risk of toxicity is minimized. Furthermore, this method is notably versatile, enabling the formation of inter- and intra-crosslinked nanogels, and is tunable through parameters such as temperature, polymer concentration, molecular weight, and radiation dose. During radiolysis of water, the formation of hydroxyl radicals leads to the generation of stable free radicals along the polymer chain, which then promote efficient crosslinking [20,21].

PVP is widely used in biomedical applications due to its minimal toxicity, exceptional biocompatibility with living tissue, and non-allergic properties [22,23]. While several reports in the literature detail the use of radiation to crosslink PVP for nanogel synthesis and property studies [24,25], there is a notable lack of systematic research exploring how polymer properties (such as molecular weight and concentration) specifically influence the characteristics of Gd-loaded PVP nanogels prepared via radiation-induced crosslinking.

Therefore, this study aimed to synthesize Gd-loaded PVP nanogels using Low-Linear-Energy-Transfer (LET) radiation from an electron beam accelerator and to systematically compare how the molecular weight and concentration of PVP affect the resulting size of these nanogels.

## 2. Materials and Methods

PVP (1300 and 360 kDa) was obtained from Sigma-Aldrich (Sigma-Aldrich, Saint Louis, MO, USA) and used without further purification. Gadolinium nitrate, Gd(NO_3_)_3_·6H_2_O, was purchased from Sigma-Aldrich. The PVP-Gd complex solutions were prepared at concentrations of 0.1 wt% and 0.01 wt%. Appropriate amounts of PVP and Gd were weighed into a 100 mL flask and filled with ultra-purified water from an EMD Millipore Direct-Q 3UV (Millipore, Bedford, MA, USA). The solutions were magnetically stirred overnight, followed by filtration with a 0.45 µm pore size syringe filter (Microliter Analytical Supplies, Inc., Suwanee, GA, USA).

### 2.1. Polymer-Gd Complex Study

The formation of the complex between PVP and Gd before irradiation was studied using several techniques for different [VP]/[Gd] ratios (R) ranging from 50 to 2.

### 2.2. Measurements

UV analysis was performed on an Analytikjena SPECORD/200 spectrophotometer (Analytikjena, Jena, Germany) using a quartz cell.

The fluorescence measurements were recorded at room temperature on the ISS PC1 spectrofluorometer (ISS Inc., Champaign, IL, USA) apparatus with a xenon lamp. The excitation and emission spectra were recorded with a slit width of 4 nm.

Fourier-Transform Infrared (FTIR) analyses were performed using the Attenuated Total Reflectance (ATR) technique with a Nicolet iS50 Analytical Flex FTIR spectrometer (Thermo Fisher Scientific, Waltham, MA, USA). Spectra were recorded over the range of 4000 to 400 cm^−1^, with a resolution of 2 cm^−1^, and an average of 64 scans per sample.

Before conducting dynamic light scattering (DLS) and zeta potential measurements, the samples were passed through a 0.45 μm filter. The analysis was performed using a Zetasizer Nano-ZS (Malvern Instruments, Malvern, UK). At least three samples were measured to determine particle size, and the mean value was reported. The temperature was maintained at 25 °C throughout the measurements. The scattering angle was fixed at 90°.

SEM/EDS measurements were conducted using the Hitachi SU-70 FEG SEM (Hitachi, Tokyo, Japan) for ultra-high-resolution morphological imaging and precise qualitative and quantitative composition analysis. SEM analysis was used to elucidate the surface morphology. Before SEM-EDS analysis, the samples were coated with a thin layer of gold and palladium to improve electrical conductivity and enhance image quality.

X-ray photoelectron spectroscopy (XPS) was performed using a Kratos Axis 165 Photoelectron Spectrometer (Kratos, Manchester, UK). The system is equipped with a monochromatic Al X-ray source, as well as a dual Al/Mg source.

For sample lyophilization, the hydrogels were thoroughly dried using a Free Zone 6 Liter −84 °C Console Freeze Dryer (Kansas City, MO, USA) operated for 72 h.

### 2.3. Ionizing Radiation

Each sample was transferred into 10 mL glass vials equipped with a cap and silicone septum. Then, N_2_O gas was introduced to convert the hydrated electron (e_aq_^−^) into hydroxyl radicals (•OH). The sample was exposed to electron beam radiation using the Medical-Industrial Radiation Facility (MIRF) at the National Institute of Standards and Technology (NIST). The electron energy of the instrument is 11 MeV with a pulse width of 6 µs and a pulse repetition rate of 120 pulses per second. Samples were placed on a temperature-controlled stage facing the electron emission port. In most of these experiments, the dose was maintained at 30 kGy, and the dose rate was 560 kGy/h. Radiochromic film dosimetry was used for the electron beam calibration. These films react to ionizing radiation through dye activation, resulting in a colorimetric change. The energy absorbed within the films was quantified using the Thermo Scientific GENESYS 20 Visible Spectrophotometer. Dosimetry is affected by three primary factors: the geometry of the sample, the type of sample, and the configuration of the experimental setup.

## 3. Results

### 3.1. PVP-Gd Mixture Prior to Irradiation 

Prior to irradiation, samples were characterized using several techniques to evaluate the complexation between PVP (0.1 wt%) and Gd nitrate in an aqueous solution at varying Gd concentrations. The aqueous solutions were characterized using DLS, UV–visible spectroscopy, and fluorescence. The FTIR, XPS, and SEM-EDS techniques were used after the lyophilization of several formulations. The zeta potential was measured as a function of the monomer (VP)/Gd ratio (R) to assess the surrounding electrical potential of PVP solutions.

Figure 1 shows the zeta potentials of PVP (0.1 wt%) solutions for both molecular weights, 1300 and 360 kDa.

Before irradiation, the PVP and PVP-Gd polymer solutions exhibited an overall negative zeta potential. As the concentration of Gd increased, the magnitude of the negative charge decreased. This reduction in zeta potential is attributed to the addition of positively charged Gd^3+^ ions.

It is observed that the solution’s pH slightly decreases when Gd^3+^ is present, as negative charges are substituted with positive ones.

Figure 2 illustrates the evolution of particle size, as determined by dynamic light scattering, in response to variations in the VP/Gd ratio, while maintaining a constant polymer concentration. In the absence of Gd^3+^ ions, at pH 6.6, the PVP solution presented small aggregates (Z-average diameter equal to 50 (±1.1) nm for PVP 1300 kDa and 47 nm (±2.2) for PVP 360 kDa; see Figure 2a,b). A slight decrease in the average size of PVP coils was observed by further addition of Gd^3+^. For a high ratio (R = 2), the size decreased from 50 nm to 40 nm (±3) for PVP 1300 kDa and from 47 nm to 45 nm (±2.5) for PVP 360 kDa. This implies that Gd ions likely coordinate with the PVP chains, resulting in a gradual reduction in large polymer aggregates and the formation of smaller colloids. It has been reported that the complexation of Gd with PVP copolymerized with negatively charged polyelectrolytes like PAA leads to the spontaneous formation of nanohydrogels exhibiting a diameter of 25 nm [16]. In that case, the Gd ions with a positive charge interact with the negative charge of PAA, forming smaller nanogels. In contrast to our case, the size of the colloids is twice as significant due to the absence of a negative charge.

The PVP-Gd aqueous solutions were also characterized using a UV–visible spectrophotometer to confirm the loading of Gd within the PVP aggregates. Figure 3a,b show UV spectra of PVP-Gd solutions in the 250–350 nm range. The absorbance band at 300 nm refers to the absorption of the Gd-PVP complex. As the concentration of the complex increases, the intensity of this band also increases [26]. To gain valuable insights into the PVP-Gd complexation, a precise method detailed later in the manuscript was used to quantify the amount of free Gd in the polymer aqueous solution.

The excitation and emission fluorescence spectra of the PVP-Gd aqueous solutions at room temperature were investigated and are summarized in Figure 4. The prepared PVP exhibited two excitation sites at 247 and 284 nm and one emission site at 477 nm. The excitation at 247 nm could be attributed to the presence of C=O, while the excitation at 284 nm could correspond to the presence of N-C in the PVP, similarly to in another report [27]. As shown in Figure 4a,b, increasing the concentration of Gd decreases the intensity of both the absorption and emission bands. Additionally, a slight shift in the PVP emission band is observed. The remarkable changes in the fluorescence spectra upon the addition of Gd suggest a significant interaction between Gd ions and the polymer. This interaction likely alters the electronic environment within the polymer, affecting the fluorescence properties and leading to the observed spectral shifts.

The aqueous solutions of PVP and PVP/Gd were lyophilized to produce a powder for additional characterization. The ATR-FTIR spectra of PVP-Gd, taken as a powder at various R values, are presented in Figure 5.

For the pure PVP spectrum, the characteristic bond observed at 3454 cm^−1^ is attributed to the O-H stretch, while the characteristic bands at 2934 cm^–1^ with two shoulders at 2952 cm^−1^ and 2886 cm^−1^ are assigned to C–H aliphatic stretching. The strong absorption peak at 1652 cm^−1^ is assigned to the carbonyl group (C=O). The peaks at 1463 and 1423 cm^−1^ are attributed to the C-N stretching in the pyrrolidone ring. The peak at 1284 cm^−1^ is assigned to the stretching of CH_2_(CH)NCO [28,29,30]. The Gd spectrum shows the characteristic bands at 733 cm^−1^ and 818 cm^−1^ related to the stretching and vibration of Gd-OH and three characteristic bands at 1031 cm^−1^, 1315 cm^−1^, and 1444 cm^−1^ assigned to nitrate vibration and stretching.

Further addition of Gd^3+^ ions leads to a noticeable shift in the wavenumber of the carbonyl group band from 1642 cm^−1^, as shown in Figure 5b. This shift is attributed to the complexation between the Gd ions and the carbonyl groups in the PVP chains, indicating strong coordination interactions. The interaction likely involves the lone pairs on the oxygen atoms of the carbonyl groups binding with the Gd^3+^ ions, which causes a change in the vibrational energy of the carbonyl bond, thereby shifting the wavenumber [16].

To gain deeper insights into the formation of PVP-Gd complexes, XPS analysis was employed to examine the surface chemical properties of both pure PVP and PVP-Gd with a Gd concentration ratio of R = 2 (indicating a high concentration of Gd). XPS provides detailed information on the binding energies of core electrons, allowing for the identification of specific chemical states and the nature of the interactions between the PVP polymer and Gd ions. By analyzing the shifts in binding energy and the appearance of new peaks, XPS can reveal the extent of complexation between the Gd ions and functional groups in PVP, such as carbonyl or nitrogen atoms, confirming the formation of stable PVP-Gd complexes.

Figure 6 shows the XPS spectra of $0PVP_{0.1}^{1300}$ (without Gd) and $2PVP_{0.1}^{1300}$ (R = 2) after lyophilization. The XPS spectrum of PVP in the presence of Gd with a ratio R = 2 (Figure 6a) indicates the presence of Gd, C, N, and O atoms in the resulting Gd-PVP.

The presence of Gd in the PVP-Gd (R = 2) samples was conclusively confirmed through XPS analysis by detecting characteristic binding energy peaks at 141 eV and 1187 eV, as shown in Figure 6a. These binding energies correspond to the Gd 4d_5_/_2_ and Gd 3d_5_/_2_ core levels, a well-known signature of gadolinium in its oxidized state [31].

The appearance of this peak not only verifies the successful incorporation of Gd into the PVP structure but also provides insights into the oxidation state and coordination environment of the Gd ions. The specific binding energy of 141 eV suggests that the Gd ions are likely in a +3 oxidation state, which is typical for gadolinium in coordination complexes.

The high-resolution spectra of the N1s core region (from 395 to 410 eV) are shown in Figure 6b. The spectrum reveals two distinct peaks at binding energies of 399 eV and 407 eV. The peak at 399 eV is attributed to the N-C bonding environment, specifically related to the nitrogen atoms within the PVP polymer backbone bonding to carbon, indicating the presence of nitrogen in a stable, covalent bond with carbon. The second peak, observed at 407 eV, is ascribed to the presence of nitrate (NO_3_^−^) species. The presence of the NO_3_^−^ peak indicates that the Gd^3+^ ions may be coordinated with residual nitrate in the sample because of the Gd precursor used during the synthesis.

The N1s spectrum, as illustrated in Figure 6b, reveals the emergence of a new peak at 402 eV. This peak is attributed to the possible coordination of nitrogen atoms within the PVP polymer with Gd ions. This novel peak indicates a significant change in the chemical environment of the nitrogen atoms, suggesting that these atoms are directly involved in binding with the Gd^3+^ ions. This interaction likely implies the nitrogen lone pairs, which coordinate with the positively charged Gd^3+^ ions, forming a stable complex.

This new peak indicates the formation of PVP-Gd complexes, where the nitrogen atoms play a key role in stabilizing the Gd ions within the polymer matrix. Comparing the intensity of this peak with that of the other peaks in the N1s spectrum can provide further insight into the extent of this coordination. A small peak at 402 eV would suggest a lower degree of coordination, indicating that a small portion of the nitrogen atoms are involved in complexation with Gd.

The O1s spectrum, shown in Figure 6c, displays a primary peak at 531 eV, corresponding to the oxygen atoms in the PVP polymer, specifically related to the carbonyl. In addition to this peak, a new peak appears at 533 eV, indicating possible coordination between oxygen atoms and Gd^3+^ ions. This novel peak at 533 eV suggests that the oxygen atoms are involved in forming complexes with Gd^3+^, leading to a shift in their binding energy. The increase in binding energy to 533 eV implies that the oxygen atoms are experiencing a different electronic environment due to their interaction with Gd. This shift can be attributed to the coordination of the lone pairs of electrons on the oxygen atoms with the Gd^3+^ ions, which influences the binding energy of the oxygen core levels. This more pronounced peak at 533 eV suggests a higher degree of coordination, indicating that a significant portion of the oxygen atoms is involved in complexation with Gd. These findings align with those previously reported, suggesting that the coordination of PVP chains with gadolinium oxide occurs preferentially through the oxygen atoms in the PVP coils [29].

As shown in the C1s spectrum depicted in Figure 6d, the spectrum of the PVP-Gd sample (R = 2) reveals three distinct peaks at 285 eV, 286 eV, and 288 eV, indicating different chemical bonding states for the carbon atoms. The peak observed at 285 eV corresponds to carbon atoms involved in C-C and C-H bonds, typically found in aliphatic hydrocarbon chains. The peak at 286 eV is associated with carbon atoms bonded to nitrogen (C-N). These bonds are characteristic of the polymer backbone in PVP. The peak at 288 eV represents carbon atoms involved in carbonyl groups (O=C-N), indicating the carbonyl groups (C=O) linked to the nitrogen atoms in the polymer chain.

As illustrated in Figure 6e, the spectrum for Gd distinctly shows the characteristic peaks corresponding to the Gd 4d core levels. Specifically, two prominent peaks are observed at 142 eV and 148.5 eV, which are attributed to the Gd 4d_5_/_2_ and Gd 4d_3_/_2_ orbitals, respectively. These peaks indicate the Gd^3+^ oxidation state, confirming the presence of gadolinium ions in their trivalent form within the PVP-Gd (R = 2) sample. The Gd 4d_5_/_2_ peak at 142 eV and the Gd 4d_3_/_2_ peak at 148.5 eV are well-established signatures of Gd^3+^ ions and demonstrate the successful incorporation of Gd into the PVP matrix. The appearance of these peaks not only validates the presence of Gd but also ensures that the gadolinium ions are in the expected oxidation state, which is crucial for their intended applications, in imaging materials. Overall, the clear observation of these Gd 4d peaks confirms that Gd has been successfully loaded into the PVP matrix, as intended when preparing the PVP-Gd (R = 2) sample [32,33].

The lyophilized solutions of $0PVP_{0.1}^{1300}$ and $2PVP_{0.1}^{1300}$ were analyzed using Scanning Electron Microscopy (SEM) and Energy-Dispersive X-ray Spectroscopy (EDS). The morphology and elemental composition of the prepared samples are presented in Figure 7. The SEM micrographs with different magnifications (Figure 7a–e) revealed distinct morphological differences between the analyzed samples. Notably, the surface morphology of the pure PVP sample was significantly altered after coordination with Gd(NO_3_)_3_. In the case of Gd-doped samples, bead-like structures were observed.

The EDS spectra further confirmed these observations. The spectrum of the $0PVP_{0.1}^{1300}$ sample (Figure 7c) showed peaks corresponding exclusively to carbon (C), nitrogen (N), and oxygen (O) atoms. In contrast, the EDS-SEM microanalysis of the $2PVP_{0.1}^{1300}$ sample qualitatively confirmed the presence of Gd, as indicated by two characteristic peaks in Figure 7f. The spectrum of $2PVP_{0.1}^{1300}$ contains C and O signals at the same positions as in the pure PVP but with reduced intensities.

Additionally, values shown in Figure 7f highlight a slight compositional variation in the Gd-doped PVP compared to pure PVP. This variation is likely due to the coordination between Gd atoms and the nitrogen and oxygen atoms present in the PVP. Similar coordination behavior has been reported in other polymer–metal systems, where rare earth elements interact with polymer matrices, affecting both the resulting materials’ chemical structure and physical properties [34,35,36]. Another possible explanation for the changes in C:N:O percentages is the presence of nitrate counterions.

### 3.2. Synthesis of Nanohydrogels Using Ionizing Radiation

The PVP solutions, with molecular weights of 1300 kDa and 360 kDa, were prepared at a concentration of 0.1 wt% and subjected to electron beam (EB) irradiation to induce polymer modification. Before irradiation, each solution was saturated with nitrous oxide (N_2_O) to enhance the production of hydroxyl radicals during the irradiation process via the reaction between hydrated electrons and N_2_O.


(1)
$$N_{2}O\;+\;{e_{aq}}^{-}\;+\;H_{2}O\;\to \;•OH\;+\;OH^{-}\;+\;N_{2}$$


The irradiation dose used was 30 kGy, and the dose rate was 560 kGy/h.

When a water-soluble polymer is exposed to ionizing radiation, it typically undergoes a crosslinking process that can lead to the formation of nanogels. The chemical reactions triggered by ionizing radiation in dilute aqueous solutions mainly involve the radiolysis of water, leading to the formation of reactive radical species and molecular products such as e_aq_−, •OH, H, H_2_, H_2_O_2,_ and H_3_O^+^.

The hydroxyl radicals produced by radiolysis are highly reactive and quickly interact with the polymer chains.


(2)
$$•OH\;+\;PVP\;\to \;H\;O\;+\;[PVP-H^{•}]$$


One of the primary reactions involves the abstraction of hydrogen atoms from the polymer chains. This reaction is shown in Figure 8.

For the successful production of nanogels, intramolecular crosslinking within the polymer chains must be effectively achieved and must dominate over intermolecular crosslinking. This intramolecular recombination depends mainly on the probability of one radical combining with another on the same polymer chain. To achieve this condition, the polymer coils must be well separated from each other, and the number of radicals generated on each chain must be maximized. These conditions are met by selecting low concentrations of a high-molecular-weight polymer and using high dose rates in the radiation experiments. The mechanism of nano- and microhydrogel formation is given in Figure 9.

After irradiating PVP (1300 kDa) solutions free of Gd, the resulting solution remained homogeneous, with a measured particle size of approximately 35 nm as determined by dynamic light scattering (DLS) [shown in Table 1]. This suggests that without Gd, the polymer chains formed stable nanogels or particles of uniform size upon irradiation.

In contrast, incorporating Gd into the PVP solutions resulted in noticeable phase separation following irradiation, as illustrated in Figure 10. The presence of Gd disrupts the homogeneous distribution of the polymer, forming distinct phases or aggregates within the solution. This phase separation suggests that the Gd ions significantly alter the crosslinking behavior or interaction dynamics of the PVP chains during the irradiation process. The phase separation phenomenon was also observed in PVP solutions with a lower molecular weight of 360 kDa. This suggests that the Gd-induced phase separation is not solely dependent on the molecular weight of the polymer but instead on the interaction between Gd and the polymer chains. These findings suggest that for a high PVP concentration, the interactions of Gd ions with the polymer chains disrupt the homogeneous crosslinking network, leading to the formation of larger aggregates and subsequent phase separation.

Under the same conditions, a concentration of 0.01 wt% was recommended for PVP solutions to prevent phase separation. Visual inspection of the irradiated solutions (Figure 11) reveals no evidence of phase separation at low Gd concentrations. However, phase separation becomes apparent at R = 10. These results indicate that at high PVP concentrations, Gd atoms can coordinate with the oxygen and nitrogen atoms of different PVP coils in the solution. In contrast, at low PVP concentrations, Gd atoms tend to coordinate within the same coil, leading to more localized crosslinking and a homogeneous solution with well-defined nanogels.

The results suggest that high PVP concentrations favor inter-chain coordination, while low concentrations favor intra-chain coordination.

The same phenomenon was observed in the case of PVP 360 kDa, demonstrating that polymer concentration is the main factor influencing the coordination and resulting properties of the Gd-PVP system.

To expand our study and gain a comprehensive understanding of controlling the PVP nanogelation process through radiation, we investigated the effects of polymer molecular weight, concentration, and the ratio of VP monomer concentration to Gd on the size of the obtained nanogels. The results are gathered in Table 1.

The initial observation from the data presented in Table 1 is the noticeable reduction in the size of the nanogels following electron beam (EB) irradiation. This decrease in size can be attributed to the high dose rate of EB, which induces significant intra-chain crosslinking within the polymer. The high energy of the electron beam facilitates the formation of covalent bonds between different segments of the same polymer chain, leading to a more compact and densely interconnected structure. As these covalent bonds form, they effectively tighten and reinforce the polymer network, reducing the overall size of the nanogels. This compacting effect creates a more rigid and stable three-dimensional network within each polymer chain, thereby decreasing the nanogel’s size while enhancing its structural integrity.

Several authors have obtained similar results, indicating a decrease in the radius of gyration of individual polymer chains during electron beam (EB) irradiation for diluted and semi-dilute polymer solutions [37,38].

The effect of the molecular weight of the starting polymer on the sizes of nanogels can also be noticed by comparing $0PVP_{0.1}^{1300}$ with $0PVP_{0.1}^{360}$; the low molecular weight leads to the formation of small nanogels. In the case of $0PVP_{0.1}^{1300}$, the energy provided by EB irradiation leads to the formation of a high number of radicals within the polymer coil; the formed radicals react with each other to generate densely crosslinked nanogels with a size of 35 nm. This increase in the size of nanogels with an increase in the molecular weight is consistent with the results of previous work [37].

The results presented in Table 1 provide clear evidence that the formation of nanogels through a radiation-induced process, in the absence of gadolinium (Gd), is highly influenced by the concentration of the polymer used. Specifically, as the concentration of PVP increases, a corresponding and significant increase in the size of the nanogels is observed.

For instance, the data shows that when PVP with a molecular weight of 1300 kDa is used, increasing the concentration from 0.01 wt% to 0.1 wt% results in a noticeable enlargement in nanogel size. A similar trend is observed with PVP of 360 kDa, where the nanogel size consistently grows as the polymer concentration is elevated within the same range. This increase in size can be attributed to the greater availability of polymer chains at higher concentrations, which likely promotes more extensive crosslinking during the radiation process, generating a larger nanogel structure. The third column of Table 1 specifically highlights this progression, where each increment in polymer concentration correlates with a proportional increase in nanogel size, reflecting the critical role of concentration in controlling the final dimensions of these nanogels.

In this investigation, the effect of temperature on the formation of nanogels was thoroughly examined. The results indicate that as the temperature increases, the size of the nanogels decreases, leading to the formation of smaller nanogel structures. This trend is particularly evident when comparing nanogels formed at ambient temperatures with those formed under elevated thermal conditions.

The observed phenomenon can be explained by the behavior of PVP chains under irradiation at higher temperatures. It has been reported by An et al. that the irradiation of PVP at high temperatures produces shrunken PVP chains, which favor the intramolecular crosslinking reaction [39].

The underlying mechanism involves the shortening of intra-radical distances on the PVP chains as the temperature increases during irradiation. At higher temperatures, the thermal energy causes the polymer chains to coil more tightly, reducing the space between reactive sites along the chains. This proximity favors intramolecular interactions, leading to the formation of smaller nanogels.

To obtain more information on the PVP-Gd complex formation and to avoid phase separation and the formation of large aggregates in the presence of Gd, Gd(NO_3_)_3_ 6H_2_O was added to nanogels after irradiation.

In the case of the PVP solution with a molecular weight of 1300 kDa and a concentration of 0.1 wt% (referred to as $0PVP_{0.1}^{1300}$), when Gd was added at a ratio R = 50, phase separation was observed, indicating the formation of large aggregates. This suggests that at higher polymer concentrations, the interaction between PVP and Gd causes instability in the nanogels formed, probably due to excessive coordination between Gd and the nanogels, which leads to the formation of dense networks that precipitate out of the solution.

In contrast, for PVP solutions with a lower concentration of 0.01 wt%, the addition of Gd at the same molar ratio (R = 50) and the ratio R = 20 resulted in homogeneous solutions, with no visible phase separation. This indicates that at lower polymer concentrations, the PVP chains are more dispersed, allowing for a more controlled interaction with Gd, which prevents aggregation and maintains solution stability.

Furthermore, measurements of the average size of the nanogels revealed that the addition of Gd caused an increase in nanogel size for both molar ratios (R = 50 and R = 20) for a PVP concentration of 0.01 wt%, regardless of the molecular weight used. This size increase could be attributed to the complexation between Gd ions and the PVP chains, which likely leads to additional physical crosslinking or expansion of the nanogels, resulting in larger structures.

With Gd concentrations ranging from 0.017 to 4.4 mmol/L, the nanohydrogels synthesized in this study show great potential as contrast agents for MRI. Our results are consistent with those of previous research, indicating that encapsulating Gd in nanogels significantly improves relaxivity, even at low concentrations, compared to free Gd complexes.

At a Gd concentration of approximately 0.9 mmol/L, Carniato, F et al. [40] developed chitosan–hyaluronic acid nanohydrogels containing Gd chelates, achieving a remarkable relaxivity of 78.0 mM^−1^ s^−1^ (at 20 MHz and 298 K), which is nearly 20 times higher than that of the free complex. This was attributed to restricted rotational motion and improved accessibility to water. Similarly, Sun et al. [19] reported gadolinium-loaded poly(N-vinylcaprolactam) (PVCL) nanogels with values (r_1_) almost twice those of free Gd-DTPA, up to 7.1 mM^−1^·s^−1^ (at 1.5 T and 37 °C). Moreau et al. [41] discovered that GdDOTA encapsulated in chitosan-based nanohydrogels exhibited better relaxivity than free GdDOTA, with values (r_1_) reaching up to 80 mM^−1^·s^−1^ (between 25 and 30 MHz).

Collectively, this research demonstrates that small amounts of Gd contained in polymer nanohydrogels can provide extremely high MRI contrast, with values (r_1_) well above those of traditional small-molecule Gd chelates.

### 3.3. Quantification of Free Gd

After the synthesis of PVP nanogels loaded with Gd, it is essential to accurately quantify the amount of free gadolinium present in the PVP solutions before and after irradiation. One technique for quantifying free gadolinium is the use of xylenol orange. Xylenol orange can selectively bind with free gadolinium ions, forming a complex that can be detected and measured spectrophotometrically. The quantification of free gadolinium using xylenol orange is a valuable method that enables the precise measurement of the amount of free gadolinium present in a sample, providing helpful information about its potential health risks and optimizing imaging protocols. A xylenol orange disodium salt (3,3′-bis[N,N-bis (carboxymethyl)aminomethyl]-o-cresol sulfone phthalein tetrasodium salt), an acetic acid buffer (pH 5.8), and a calibration curve were prepared as previously reported by Barge et al. [42]. The absorbance of free Gd^3+^ leakage was measured at wavelengths of 433 nm and 573 nm. After coordination with gadolinium ions, the relative intensities of the dye absorption bands change: the first band decreases while the second band increases (Figure 12). The concentration of free Gd^3+^ was calculated using Equation (3):
(3)$$Y\;=\;A\;+\;B\left[{Gd}^{3+}\right]$$
where Y is the ratio of the absorbances at 573 nm to those at 433 nm.

The determination of leached Gd^3+^ ions was performed using 2 mL of 0.001 M xylenol orange solution in acetate buffer (pH 5.81) and 200 μL of supernatant of Gd^3+^ dispersed in PVP solutions before and after irradiation.

Figure 13 shows the percentage of free Gd in PVP (1300 kDa, 360 kDa) solutions before and after irradiation for two ratios, R = 50 and R = 20.

As shown in Figure 13, the concentration of free Gd in the PVP solution increases after irradiation. This outcome may be attributed to the leakage of Gd ions from the PVP coils following the chemical crosslinking of the PVP chains. It is important to highlight that the concentration of free gadolinium (Gd) in the PVP sample with a molecular weight of 360 kDa is significantly higher than that in the sample with a molecular weight of 1300 kDa before irradiation, as shown in the data from Figure 13.

These findings suggest that a higher-molecular-weight PVP (1300 kDa) provides more binding sites, resulting in more efficient Gd coordination and lower free Gd concentrations. In contrast, lower-molecular-weight PVP (360 kDa) has fewer available binding sites, resulting in higher free Gd levels.

To prevent the leakage of gadolinium (Gd) from the PVP chains after the irradiation process, a specific strategy was employed: Gd(NO_3_)_3_ 6H2O was added to the nanogel solution after irradiation had been completed. This method aimed to ensure that Gd ions would be more effectively retained within the PVP nanogel structure, minimizing any potential release of free Gd into the surrounding medium.

The quantification of free Gd in the supernatant, as depicted in Figure 14, showed that when Gd was added to the nanogel solution after irradiation, there was a slight but noticeable decrease in the concentration of free Gd. This decrease indicates that more Gd ions were successfully coordinated with the PVP nanogels, reducing the amount of Gd remaining unbound and free in the solution. The addition of Gd after irradiation improves the stability of the Gd-PVP complex by binding Gd to already-crosslinked and rigid PVP chains, thereby reducing the risk of Gd leaching from the nanogels. In contrast, adding Gd before irradiation may result in less-secure binding, as some Gd ions might not fully integrate into the forming nanogel structure.

These findings indicate that adding Gd after irradiation is a promising strategy to minimize the concentration of free Gd, thereby enhancing the stability of the nanogel and reducing the potential for Gd-related toxicity. In that case, the ions are more securely bound to the PVP nanogels, effectively preventing their leakage into the surrounding medium. This approach is particularly beneficial for biomedical applications where Gd retention is critical for safety and efficacy. However, while this method reduces free Gd and its associated toxicity, it also leads to the formation of larger nanogels, as shown in Table 1. The increase in nanogel size is likely due to the additional crosslinking or aggregation induced by the introduction of Gd ions after the nanogels have already formed.

Studies have shown that free gadolinium (Gd) ion concentrations exceeding 0.1 mmol/L can be harmful, potentially leading to toxicity and adverse effects in biological systems [43]. For the PVP nanogels developed in this study, the concentration of free Gd is well below this harmful threshold. Specifically, for both VP-to-Gd molar ratios examined (R = 50 and R = 20), the free Gd concentration was measured at less than 0.1 mmol/L.

It is important to note that free Gd^3+^ ions can cause inaccuracies in MRI sensitivity, because unbound Gd^3+^ can produce inconsistent contrast, distorting the quality of the MRI images. To prevent this, minimizing free Gd^3+^ ions in nanogel solutions is crucial before any MRI experiments. One effective method to achieve this is using a chelating agent such as diethylenetriaminepentaacetic acid (DTPA).

## 4. Conclusions

In this study, gadolinium (Gd)-loaded PVP nanogels were synthesized using electron beam irradiation. Characterization techniques, including FTIR, fluorescence, and UV–visible spectroscopy, confirmed effective complexation between Gd and PVP. Dynamic light scattering (DLS) measurements revealed that the size of the nanogels is significantly influenced by several parameters, including the molecular weight of PVP, polymer concentration, the VP-to-Gd ratio, and the timing of Gd addition relative to the irradiation process. Additionally, the quantification of free Gd demonstrated how variations in molecular weight and the timing of Gd addition (before or after irradiation) affect the concentration of free ions.

The most effective system was achieved with the formulation $50PVP_{0.01}^{1300}$, which produced small nanogels and exhibited a low concentration of free Gd. This formulation demonstrates excellent suitability for MRI applications, combining optimal nanogel size with minimal free gadolinium, which enhances imaging performance and reduces potential toxicity.

## Figures and Tables

**Figure 1 polymers-17-02100-f001:**
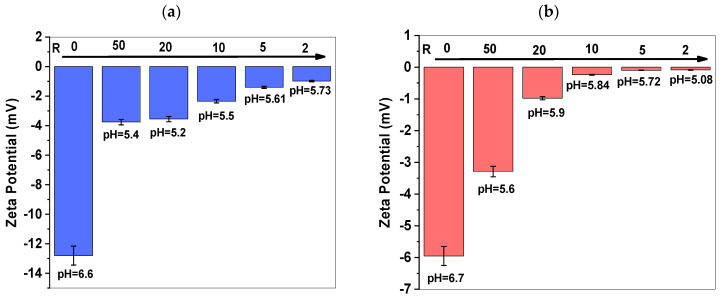
Zeta potential of PVP (0.1 wt%) solutions before irradiation: (**a**) PVP 1300 kDa and (**b**) 360 kDa for R from 50 to 2.

**Figure 2 polymers-17-02100-f002:**
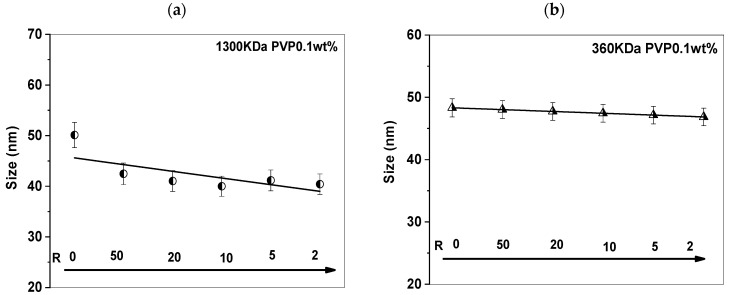
Size change in PVP (0.1 wt%) solutions before irradiation: (**a**) PVP 1300 kDa and (**b**) 360 kDa for R from 50 to 2.

**Figure 3 polymers-17-02100-f003:**
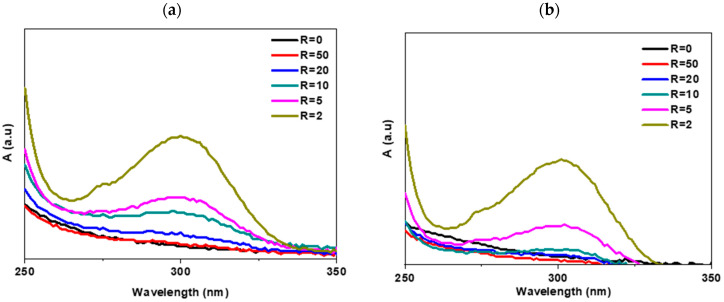
UV spectra of PVP-Gd complex in aqueous solutions prior to irradiation, where [PVP] is 0.1 wt%: (**a**) PVP 1300 kDa and (**b**) 360 kDa for R from 50 to 2.

**Figure 4 polymers-17-02100-f004:**
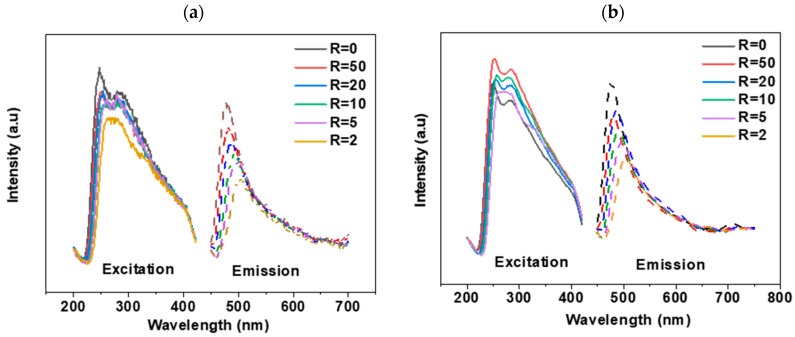
Fluorescence spectra of PVP-Gd complex aqueous solutions before irradiation, where [PVP] is 0.1 wt%: (**a**) PVP 1300 kDa and (**b**) 360 kDa for R from 50 to 2.

**Figure 5 polymers-17-02100-f005:**
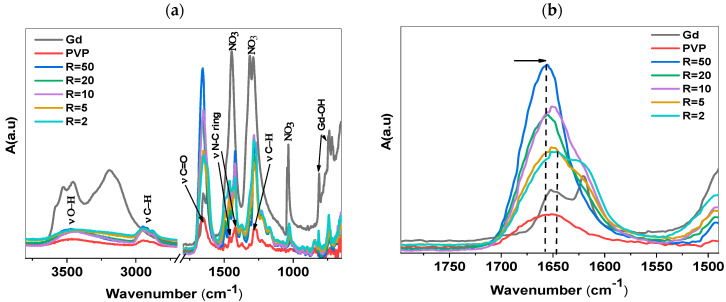
(**a**) FTIR spectra of PVP-Gd with R varying from 50 to 2 after lyophilization of PVP-Gd solutions; (**b**) zoomed-in view of C=O band range showing the shift after the addition of Gd.

**Figure 6 polymers-17-02100-f006:**
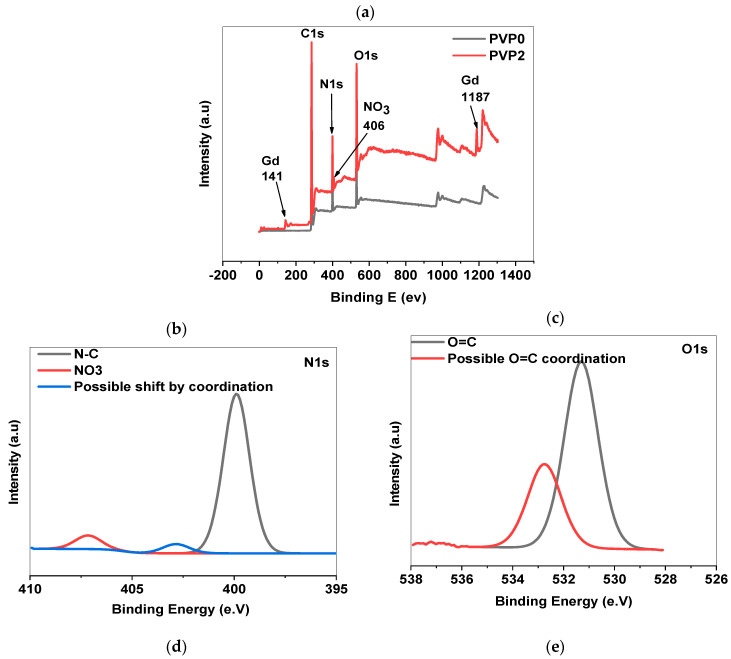
High-resolution XPS spectra of (**a**) PVP and PVP-Gd (R = 2), (**b**) N1s, (**c**) O1s, (**d**) C1s, and (**e**) Gd 4d for PVP-Gd (R = 2), respectively.

**Figure 7 polymers-17-02100-f007:**
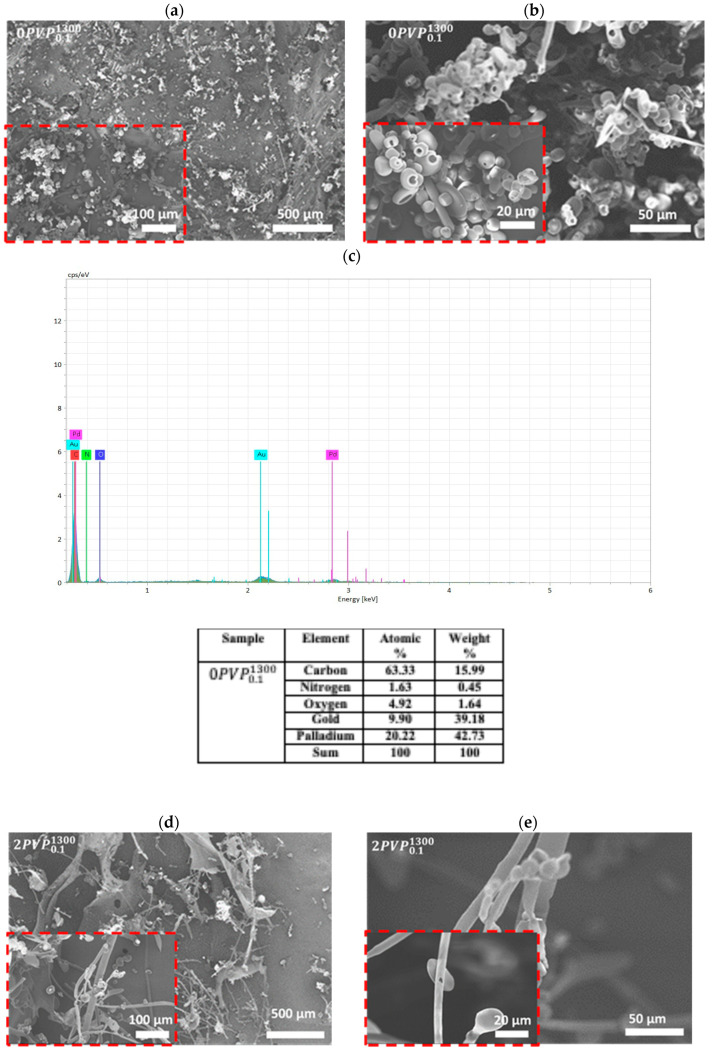
SEM of the lyophilized $0PVP_{0.1}^{1300}$ at (**a**) 500 μm and (**b**) 50 μm magnification and (**c**) the corresponding EDS elemental analysis. SEM images of the lyophilized $2PVP_{0.1}^{1300}$ hydrogels with (**d**) 500 μm and (**e**) 50 μm magnification and (**f**) the corresponding EDS elemental analysis.

**Figure 8 polymers-17-02100-f008:**
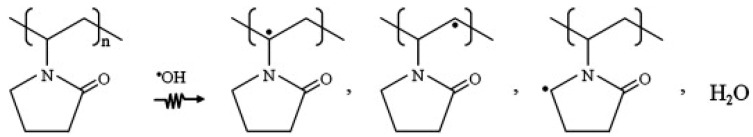
The abstraction of hydrogen atoms from the polymer’s backbone to produce PVP radicals.

**Figure 9 polymers-17-02100-f009:**
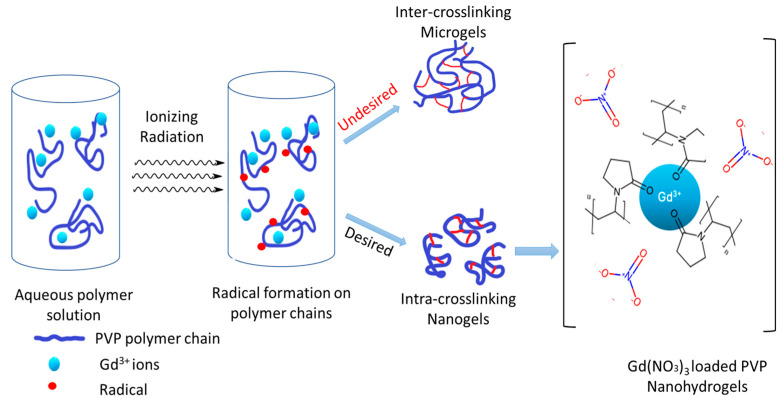
Scheme of radiation-induced synthesis of Gd-loaded PVP nano- and microhydrogels.

**Figure 10 polymers-17-02100-f010:**
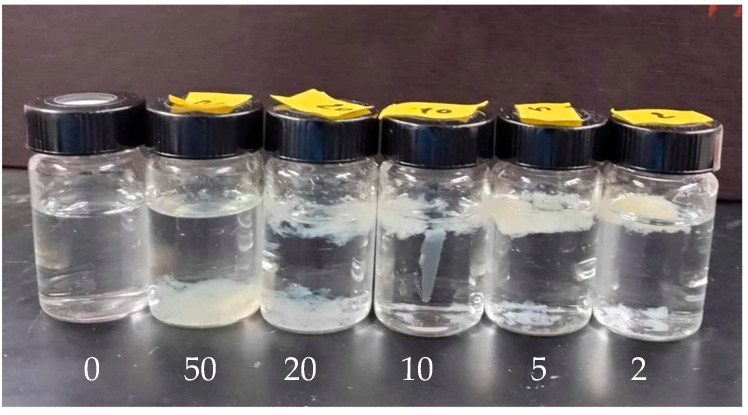
A photograph of PVP (1300 kDa, 0.1 wt%) solutions after EB irradiation in the presence of Gd with R varying from 50 to 2.

**Figure 11 polymers-17-02100-f011:**
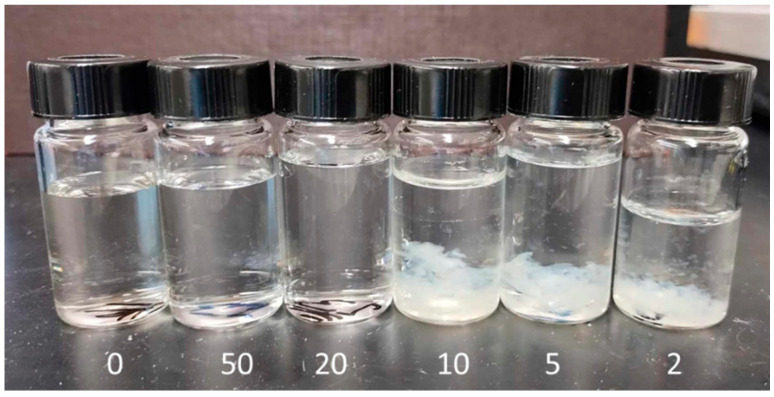
A photograph of PVP (1300 KDa, 0.01 wt%) solutions after EB irradiation in the presence of Gd with R varying from 50 to 2.

**Figure 12 polymers-17-02100-f012:**
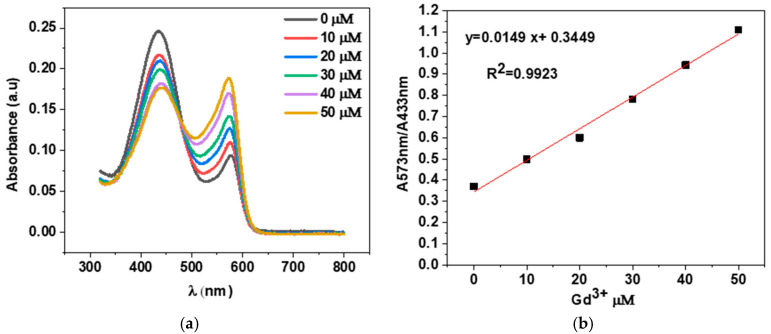
Spectrophotometric determination of free Gd ions by xylenol orange: (**a**) UV-vis spectrum of xylenol orange in acetate in the presence of varied concentrations of Gd from 0 to 50 µm; (**b**) calibration curve obtained for absorbance ratio A573/A433 versus Gd^3+^ concentration.

**Figure 13 polymers-17-02100-f013:**
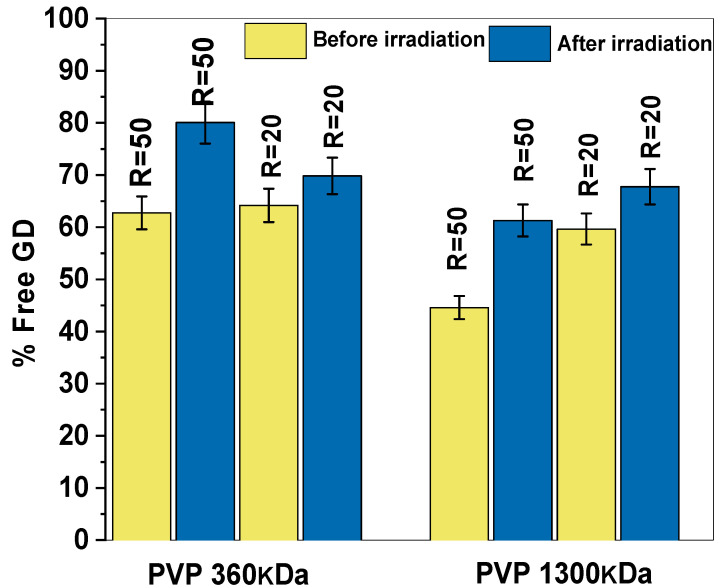
Percentages of free Gd in the PVP 0.01 wt% solutions before and after irradiation. For PVP 1300 kDa and 360 kDa.

**Figure 14 polymers-17-02100-f014:**
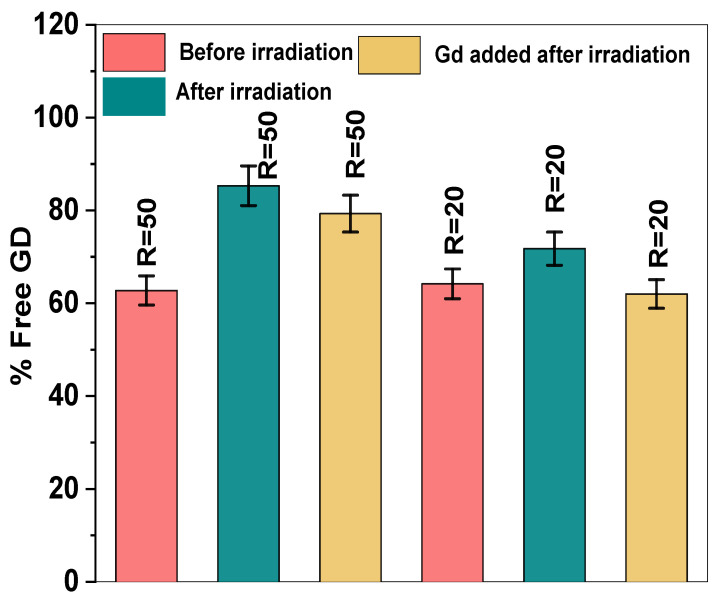
Percentage of free Gd before and after irradiation and when Gd was added to PVP (360 kDa, 0.01 wt%) nanogel solutions.

**Table 1 polymers-17-02100-t001:** Nanogels’ size for aqueous PVP solutions with two molecular weights (1300, 360 kDa) and two concentrations (0.1 wt%,0.01 wt%) before and after e-beam irradiation with a dose of 30 kGy and dose rate of 560 kGy/h.

	Unirradiated	Irradiated	Irradiated at 77 °C	Gd Added After Irradiation at Room T
	d (nm)			
$0PVP_{0.1}^{1300}$ $50PVP_{0.1}^{1300}$ $20PVP_{0.1}^{1300}$	50.11 ± 1.1	35.84 ± 0.2	-	-
	42.46 ± 3.2	-	-	-
	41.03 ± 2.4	-	-	-
$0PVP_{0.01}^{1300}$ $50PVP_{0.01}^{1300}$ $20PVP_{0.01}^{1300}$	62.855 ± 1.5	27.24 ± 0.1	24.62 ± 0.05	-
	62.02 ± 2.3	30.31 ± 0.2	26.55 ± 0.1	36.31 ± 0.2
	61.96 ± 3.1	106.90 ± 0.1	54.95 ± 0.3	40.24 ± 0.1
$0PVP_{0.1}^{360}$ $50PVP_{0.1}^{360}$ $20PVP_{0.1}^{360}$	47.77 ± 2.2	33.56 ± 0.2	-	-
	48.90 ± 3.1	-	-	-
	47.40 ± 1.2	-	-	-
$0PVP_{0.01}^{360}$ $50PVP_{0.01}^{360}$ $20PVP_{0.01}^{360}$	71.11 ± 3.1	28.53 ± 0.3	-	-
	70.00 ± 2.1	51.63 ± 0.1	-	41.33 ± 0.3
	68.84 ± 1.1	82.74 ± 0.1	-	44.21 ± 0.1

## Data Availability

The original contributions presented in this study are included in the article. Further inquiries can be directed to the corresponding authors.

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
