# Peer review of "Synthesis of Gadolinium-Loaded Poly(N-vinyl-2-pyrrolidone) Nanogels Using Pulsed Electron Beam Ionizing Irradiation"

_polymers, 2025, doi:10.3390/polym17152100_

Round 1
Reviewer 1 Report
Comments and Suggestions for Authors
The investigation of the authors' group is devoted to the study of the influence of concentration and molecular weight on the electron-beam synthesis of PVP nanogels loaded with Gd. Despite the relevance of this work, it contains shortcomings, which will be discussed below.
Comment 1. Since this study does not investigate the relaxation efficiency of the obtained compounds, it is difficult to assess whether the amount of gadolinium contained in the obtained nanogels is sufficient to obtain good contrast in MRI images. It is necessary to compare the gadolinium content with similar nanogels obtained by other researchers for whom the value of relaxation efficiency is known, or with drugs used in clinical practice, such as Magnevist or Gadovist.
Comment 2. The manuscript is poorly formatted: there are missing spaces in some places, typos, no superscripts/subscripts, periods instead of commas, incorrect spelling of chemical formulas (L.103). This creates the impression that the work is not very important to the authors. Abbreviations are not used as they should be (for example, the L.130 deciphers the concept of X-ray photoelectron spectroscopy (XPS), then the L.153 correctly use the abbreviation, and then the L.225 again uses the full name of the method.
Comment 3. Section 2.2 does not specify the brand of the spectrofluorometer. It does not specify how the IR spectra were obtained: in pellets with potassium bromide or using a FTIR attachment. For measurements using the DLS method, the temperature of the samples and the scattering angle are not specified. In the L.133 line, the “lyophilization” should be used instead of the “hydrophilization”.
Comment 4. Section 2.3 says nothing about the irradiation atmosphere; although later in the manuscript, (L.337-338) there is information that the irradiation took place in an atmosphere of nitrous oxide.
Comment 5. The results presented in Table 1 must be rounded according to the rules
Comment 6. In Figure 7, the presented EDS spectra are of very poor quality. It may be worthwhile to present the EDS data in a separate table, without presenting the spectra. In addition, it is necessary to explain where the large amount of gold and platinum comes from by adding a description of the sample preparation in the section 2.2.
Comment 7. Figure 2 needs improvement. The values ​​on the ordinate axis can start from something other than zero, and the numbers on the abscissa axis can be aligned.
Сomment 8. In fluorescence spectra (Figure 4), emission and excitation spectra should be marked with different lines (e.g. straight and dotted) and this should be noted in the legend of the graphs. In Figure 4b, in the legend, instead of PVP, there should be R. What does mean λ=210 in Figure 4A?
Сomment 9. The text of the manuscript states that the vibration band of the C=O bond of PVP in the nanogel is shifted. However, this is not obvious in Figure 5B. A straight vertical line should be drawn.
Сomment 10. What is shown in Figure 9 is not a mechanism, but rather a scheme.
Сomment 11. In Figure 12A, the ordinate axis should say "absorbance"
Сomment 12. In Figure 6E there is no ordinate axis designation.
Author Response
We express our gratitude to the reviewer for their diligent and thorough examination of our manuscript, as well as for their insightful comments and valuable suggestions. We acknowledge alignment with the majority of their remarks, prompting us to refine our manuscript Ref.: polymers-3760936: Synthesis of Gadolinium-Loaded Poly(vinylpyrrolidone) Nanogels Using Pulsed Electron Beam Ionizing Irradiation.
Below, we respond to each of the reviewer's comments. We aim to ensure our responses thoroughly address the reviewer's concerns and improve the quality of our work. Additionally, we remain open to including any further suggestions the reviewer may have to enhance the revised manuscript.
Sincerely,
We would like to inform the reviewer that the introduction, experimental section, figures, and language throughout the entire manuscript have been revised accordingly.

Reviewer 2 Report
Comments and Suggestions for Authors
Reviewer Comments for Manuscript Number polymers-3760936
This manuscript, “Synthesis of Gadolinium-Loaded Poly(vinylpyrrolidone) Nanogels Using Pulsed Electron Beam Ionizing Irradiation” deals with the fabrication of nanogels and incorporation of gadolinium. This work is discussed in detail and appears to be well-received. But I have a few suggestions for consideration.
- A particle size distribution histogram could be beneficial.
- Swelling is an important aspect of gels. Please provide the swelling profile in distilled water.
- Figure 9: How can undesirable inter-crosslinking be avoided? What is the effect of E-beam on gadolinium?
- The language of the manuscript should be improved, and grammatical errors should be addressed. A few observations are listed below:
On line 45, a new sentence starts with a lower letter. Lines 48 to 66 should be constructed in a paragraph. Lines 78 to 90 are not backed by the reference. Adjust the (cm-1) unit on Line 119. The measurements section can be arranged in a paragraph. Use proper SI units, and a consistent gap between numbers and units should be maintained. Unnecessary gaps should be removed. The numbering of equations can be added.
- MRI studies should be conducted to evaluate the true potential of the gadolinium-loaded nanogel.
- The text of Heading 3.1 is too lengthy. In my opinion, the data for before and after irradiation should be merged to give a better understanding, and subheadings should be provided for better readability.
Author Response
We express our gratitude to the reviewer for their diligent and thorough examination of our manuscript, as well as for their insightful comments and valuable suggestions. We acknowledge alignment with the majority of their remarks, prompting us to refine our manuscript Ref.: polymers-3760936: Synthesis of Gadolinium-Loaded Poly(vinylpyrrolidone) Nanogels Using Pulsed Electron Beam Ionizing Irradiation.
Below, we respond to each of the reviewer's comments, with our responses highlighted in blue. We aim to ensure our responses thoroughly address the reviewer's concerns and improve the quality of our work. Additionally, we remain open to including any further suggestions the reviewer may have to enhance the revised manuscript.
Sincerely,
We would like to inform the reviewer that the introduction, experimental section, figures, and language throughout the entire manuscript have been revised accordingly.

Round 2
Reviewer 1 Report
Comments and Suggestions for Authors
The Authors have improved and revised their work
Reviewer 2 Report
Comments and Suggestions for Authors
NA